# Partial Protection of Goats against *Haemonchus contortus* Achieved with ADP-Ribosylation Factor 1 Encapsulated in PLGA Nanoparticles

**DOI:** 10.3390/vaccines12101188

**Published:** 2024-10-18

**Authors:** Muhammad Waqqas Hasan, Javaid Ali Gadahi, Muhammad Haseeb, Qiangqiang Wang, Muhammad Ehsan, Shakeel Ahmad Lakho, Ali Haider, Tahir Aleem, Mingmin Lu, Ruofeng Yan, Xiaokai Song, Xiangrui Li, Lixin Xu

**Affiliations:** 1MOE Joint International Research Laboratory of Animal Health and Food Safety, College of Veterinary Medicine, Nanjing Agricultural University, Nanjing 210095, China; 2015207037@njau.edu.cn (M.W.H.); drgadahi@yahoo.com (J.A.G.); 2016207041@njau.edu.cn (M.H.); 2016107053@naju.edu.cn (Q.W.); mehsan124@gmail.com (M.E.); 2017207046@njau.edu.cn (S.A.L.); 2018207074@njau.edu.cn (A.H.); 2018207076@njau.edu.cn (T.A.); mingmin.lu@njau.edu.cn (M.L.); yanruofeng@njau.edu.cn (R.Y.); songxiaokai@njau.edu.cn (X.S.); lixiangrui@njau.edu.cn (X.L.); 2Key Laboratory of Molecular Target & Clinical Pharmacology and the State & NMPA Key Laboratory of Respiratory Disease, School of Pharmaceutical Science, The Fifth Affiliated Hospital, Guangzhou Medical University, Guangzhou 511436, China

**Keywords:** *H. contortus*, ARF1, PLGA polymer, nanovaccine, immunomodulation, goats

## Abstract

Background: *Haemonchus contortus* (*H. contortus*), a nematode with global prevalence, poses a major threat to the gastrointestinal health of sheep and goats. In an effort to combat this parasite, a nanovaccine was created using a recombinant ADP-ribosylation factor 1 (ARF1) antigen encapsulated within poly lactic-co-glycolic acid (PLGA). This study aimed to assess the effectiveness of this nanovaccine in providing protection against *H. contortus* infection. Methods: Fifteen goats were randomly divided into three groups. The experimental group received two doses of the PLGA encapsulated rHcARF1 (rHcARF1-PLGA) nanovaccine on days 0 and 14. Fourteen days after the second immunization, both the experimental and positive control groups were challenged with 8000 infective larvae (L3) of *H. contortus*, while the negative control group remained unvaccinated and unchallenged. At the end of the experiment on the 63rd day, all animals were humanly euthanized. Results: The results showed that the experimental group had significantly higher levels of sera IgG, IgA, and IgE antibodies, as well as increased concentrations of cytokines, such as IL-4, IL-9, IL-17, and TGF-β, compared to the negative control group after immunization. Following the L3 challenge, the experimental group exhibited a 47.5% reduction in mean eggs per gram of feces (EPG) and a 55.7% reduction in worm burden as compared to the positive control group. Conclusions: These findings indicate that the nanovaccine expressing rHcARF1 offers significant protective efficacy against *H. contortus* infection in goats. The results also suggest the need for more precise optimization of the antigen dose or a reassessment of the vaccination regimen. Additionally, the small sample size limits the statistical rigor and the broader applicability of the findings.

## 1. Introduction

*Haemonchus contortus* (*H. contortus*) ranks among the most harmful gastrointestinal nematodes, particularly affecting small ruminants, exhibiting a ubiquitous global distribution [1]. This helminthic parasite infects its host population primarily through the ingestion of forage contaminated with infective larvae, subsequently establishing residence within the abomasum to feed on the blood of its host. Clinical manifestations in afflicted animals encompass dehydration, debility, diminished production yields, and anemia [2]. Notably, young lambs are especially vulnerable to the lethal consequences of this nematode infection [3]. Unfortunately, an effective vaccine targeting *H. contortus* remains absent. As a result, the primary approaches for managing this parasitic menace entail meticulous pasture management practices and the periodic administration of anthelmintic treatments. Unfortunately, protracted reliance on anthelmintics has spurred the emergence of resistance to these drugs. Consequently, the proliferation of drug-resistant *H. contortus* strains now presents a substantial threat to the sustainability of small ruminant husbandry worldwide [4,5]. The advent of efficacious anti-parasitic vaccines stands to introduce a pivotal novel dimension to the strategic management and control of GIN infections in small ruminants.

ADP-ribosylation factor 1 (ARF1) isolated from *H. contortus*, denoted as HcARF1, was pinpointed as the primary antigen of interest within the infective larval stages [6]. ARF1 is categorized within the group of Ras-associated GTPases and plays a pivotal role in facilitating vesicular trafficking, as well as functioning as a signal transduction mediator [7,8,9,10]. ARF1 is involved in critical biological functions necessary for the parasite’s survival within its host. It likely plays a role in regulating metabolic pathways and enabling the parasite to adapt to the host environment, both of which are vital for sustaining the infection [11]. This protein has been meticulously characterized across a spectrum of eukaryotic organisms, including model species such as *Drosophila melanogaster*, *Saccharomyces cerevisiae*, *Plasmodium falciparum*, and *Caenorhabditis elegans* [12,13,14]. In pursuit of an effective vaccine design against *H. contortus*, Hasan et al. recently employed biodegradable nanoparticles, including PLGA and chitosan NPs, to encapsulate HcARF1. This approach demonstrated enhanced immune responses in murine subjects [15]. Complementing these findings, in vitro investigations have underscored the immunomodulatory properties of HcARF1, shedding light on its crucial role in the pathogenesis of *H. contortus* [16].

Within the realm of nano-medicine, the biodegradable polymer poly (lactic-co-glycolic acid) (PLGA) has emerged as a subject of heightened interest, offering a novel approach to combat infectious parasites [15,17]. PLGA, when harnessed in conjunction with specific antigens, has demonstrated its capacity to elicit a potent immune response, proving effective against various parasitic infections, including *leishmaniasis* and *toxoplasmosis* [18,19,20]. PLGA nanoparticles (NPs) play a multidimensional role, not only in optimizing vaccine delivery but also in reducing the required dosage and injection frequency and extending the antigen’s presence within the host organism [21]. Furthermore, a subset of NPs possess inherent immunomodulatory properties, capable of triggering or augmenting the immune response [22,23,24,25]. According to a recent study by Hasan et al., PLGA shows considerable efficacy in encapsulating parasitic antigens and can trigger significant immunogenic responses in the host, leading to notable lymphocyte proliferation and activation of T cells and dendritic cell (DC) subsets [17]. Our previous research demonstrated that rHcARF1 exhibits robust immunogenicity against *H. contortus* in vitro. When encapsulated within PLGA nanoparticles, this protein effectively induced protective immunity by modulating both cellular and humoral immune responses in a murine model [15,16]. PLGA particles with antigens and toll-like receptor ligands trigger strong antitumor immune responses. Encapsulating the novel dsRNA adjuvant Riboxxim in PLGA markedly boosts dendritic cell activation and CD8^+^ T cell responses over conventional dsRNA analogs. This approach effectively inhibits primary tumor growth, prevents metastases, and improves survival, with even greater efficacy when combined with immune checkpoint blockade [26].

In our present study, we are delving into the protective response elicited by an antigen delivery system, specifically the antigen–NPs complex denoted as HcARF1-PLGA, in the context of *H. contortus* infection. Our findings suggest that this nanovaccine regimen, when coupled with appropriate management practices, may offer a robust defense against the infection.

## 2. Experimental Materials and Procedures

### 2.1. Ethics Approval

All animals in this study were subject to rigorous ethical oversight and management procedures, adhering to the directives set forth by the Animal Ethics Committee at Nanjing Agricultural University, China. Furthermore, all animal experimentation strictly followed the guidelines established by the Animal Welfare Council of China. Additionally, the experimental protocols obtained official approval from the Science and Technology Agency of Jiangsu Province, as indicated by the approval ID SYXK (SU) 2010-0005.

### 2.2. Reagents, Parasites, and Proteins

PLGA (poly (lactic-co-glycolic acid)) with a composition of Lactic acid/Glycolide of 65:35 and a molecular weight range of 40,000–75,000, as well as polyvinyl alcohol (PVA) with a molecular weight range of 31,000–50,000, were procured from Sigma-Aldrich (St. Louis, MO, USA). The Micro-BCA^TM^ Protein Assay Kit was obtained from CW Biotech (Beijing, China PR). Antibodies, including rabbit anti-goat IgG, rabbit anti-goat IgA (abcam, Shanghai, China), and the Immunoglobulin E (IgE) detection kit (Jinyibai, Nanjing, China), were utilized. ELISA kits for goat IL-9, goat IL-4, goat IL-17, goat TGF-β1, and goat IFN-γ were sourced from HengYuan (Shanghai, China).

In this investigation, the *H. contortus* strain was sustained via consecutive passages within our laboratory setting. The third-stage (L3) larvae were cultivated at the Laboratory of Veterinary Parasitic Diseases at Nanjing Agricultural University [27]. Total RNA was isolated from adult worms of *H. contortus* collected from the abomasum of donor goats as described previously [28], and cDNA was synthesized by reverse transcription reaction using a cDNA Kit. The complete ORF of HcARF1 was amplified via RT-PCR using primers specific to the *H. contortus* ARF gene (GI: 533372025, GenBank accession HF964523.1). Purified PCR products were cloned into the pMD19-T vector and transformed into *E. coli* (DH5α) cells. Positive clones were confirmed through BamHI/XhoI digestion, sequenced, and analyzed using DNAssist 3.11 software. The HcARF1 gene was cloned into the pET32a (+) vector, sequenced to verify correct insertion, and expressed in *E. coli* BL21(DE3) cells with IPTG induction. The histidine-tagged fusion protein was purified using His Bind^®^ Resin, dialyzed in PBS to remove imidazole, and endotoxins were eliminated with ToxinEraser™. Purity, expression, and concentration were assessed via a 12% SDS-PAGE and Coomassie staining, and storage of the rHcARF1 protein was conducted in-house at −80 °C [16].

### 2.3. The Optimization of Polyvinyl Alcohol (PVA)

Prior to the commencement of PLGA NP synthesis, a comprehensive investigation was conducted to determine the optimal concentration of PVA. In the current investigation, we considered three distinct gradients of PVA concentrations, specifically set at 1%, 4%, and 6%. Subsequently, we proceeded to assess the characteristics of the PLGA NPs generated under these varying PVA concentrations utilizing scanning electron microscopy (SEM), employing a JEOL IT-100 microscope from Tokyo, Japan.

### 2.4. Synthesis of rHcARF1-Loaded PLGA NPs

PLGA nanoparticles (NPs) were fabricated using the double emulsion technique, as previously described [27], with some modifications to maintain sterility. In brief, the inner aqueous phase was created by dissolving the recombinant protein rHcARF1 (2.2 mg/mL) in a 6% PVA solution. Simultaneously, the organic phase was prepared by dissolving 5% PLGA in methylene chloride, resulting in a solution of 50 mg of PLGA per 1 mL of methylene chloride. The inner aqueous phase and the organic phase were amalgamated to form the water-oil (w/o) emulsion, utilizing an ultrasonic processor (JY92-IIN, NingBo Scientz Biotechnology, Ningbo, China) for a 4-min duration (40 W, 5 s, 5 s), while maintaining an ice bath. This w/o emulsion was subsequently introduced into the external aqueous phase, comprised of a 6% PVA solution in deionized water. The same sonication conditions were applied to obtain the final emulsion, forming a water-oil-water (w/o/w) configuration. The organic solvent within the emulsion was eliminated through evaporation under magnetic stirring for a duration of 4–5 h in a chemical fume cupboard, conducted at room temperature. The resultant antigen-loaded NPs (rHcARF1-PLGA) were separated from the NP solution through centrifugation at 20,000× *g* for 40 min at 4 °C. The supernatant was collected to determine protein loading efficiency using the Micro-BCA^TM^ protein assay kit. At the same time, the precipitated NPs were subjected to two wash cycles via centrifugation with ultrapure water. The NPs were subsequently placed within a freeze-drying machine (Labconco^TM^, Thermo Fisher Scientific, Waltham, MA, USA) for a period of 24 h and stored at −80 °C until they were ready for use in further experiments.

For the preparation of blank PLGA NPs, to serve as a control in subsequent experiments, the antigen rHcARF1 was intentionally omitted from the process, ensuring these NPs remained antigen-free.

### 2.5. Physical Characteristics of rHcARF1-Loaded PLGA NPs

To determine the encapsulation efficiency (EE) and loading capacity (LC) of rHcARF1 within the nanoparticles (NPs), we employed the Micro-BCA^TM^ protein assay kit. These parameters were calculated using the following equations [29,30]:EE = (total protein − unbound protein)/total protein × 100%
LC = loaded protein/total mass of the nanovaccine × 100%

The surface structure and dimensions of the PLGA NPs were examined using a cold field emission SEM (JEOL IT-100, Tokyo, Japan). Subsequently, the powdered NPs were prepared for examination by being loaded onto aluminum stubs and coated with platinum.

To validate the integrity of the loaded protein, a sodium dodecyl sulfate–polyacrylamide gel electrophoresis (SDS-PAGE) gel was run.

### 2.6. Goats Immunization

The experiment was conducted after receiving the necessary permissions from the Animal Care and Ethics Committee at Nanjing Agricultural University (Approval ID: 201009022) for the inoculation of goats with the nanovaccine. Fifteen locally crossbred female goats, aged between 5 and 6 months, were reared in a controlled indoor environment that was maintained free from nematode infestations. The animal housing and surroundings were subjected to daily cleaning procedures, with disinfection executed every three days during the trial. The goats were categorized into three groups, each consisting of five animals, with an emphasis on achieving a balanced distribution of body weight (15 ± 2.5 kg). All goats were clinically and visibly healthy before undergoing immunization. To prevent natural helminth infections, all goats received Levamisole at a dosage of 8 mg/kg of body weight every two weeks, in accordance with the recommended guidelines.

In the experimental group (n = 5), the goats received the rHcARF1-PLGA nanovaccine intramuscularly (I/M) on day 0. The nanovaccine had a rHcARF1 concentration of 2.7 ng/µL and was administered in 1 mL of PBS (pH 7.4). The inoculation dose was evenly distributed between two different injection sites (the thigh and shoulder muscles of the goats). A subsequent booster dose of the nanovaccine was administered after a 2-week interval on day 14. The negative control group (n = 5) remained unvaccinated and unexposed to L3 challenges, yet they received mock vaccinations with 1 mL of PBS. In a similar manner, the positive control group (n = 5) remained unvaccinated. They were also mock-vaccinated with 1 mL of PBS (pH 7.4). On day 28, animals from both the experimental and positive control groups were orally challenged with 8000 L3 (infective larvae of *H. contortus*). On day 64, all animals were sacrificed humanely for further investigation. Schedule for vaccinations, challenge with infection, and sampling are mentioned in Table 1.

### 2.7. Collection of the Serum Samples

Blood samples were obtained from the goats via the jugular vein at specific time points, namely 0, 14, 28, 43, 54, and 63 days throughout the experimental period. Each blood collection event involved drawing five milliliters of blood from each goat collected into sterile plain universal tubes. The samples were refrigerated overnight at 4 °C to allow coagulation of the blood. The resulting supernatant was then subjected to centrifugation at 3200× *g* for 20 min. The separated serum was subsequently preserved at −20 °C until required for analysis.

### 2.8. ELISA for the Determination of Antibodies in Sera

The levels of serum IgA and IgG were determined using the indirect enzyme-linked immunosorbent assay (ELISA) method, following established protocols [27]. In brief, 96-well clear, polystyrene high bind strip well^TM^ microplates (Corning, San Diego, CA, USA) were treated with 240 ng/µL of rHcARF1 and allowed to incubate at 4 °C for all the night.

Next, serum samples, appropriately diluted in PBS containing 0.5% Tween-20 (PBST), were introduced into every well of a 96-well plate and incubated at 37 °C for 1–2 h. Following incubation, the wells underwent thorough washing with PBST and were subsequently incubated with Horseradish Peroxidase (HRP)-conjugated rabbit anti-goat IgA and HRP-conjugated rabbit anti-goat IgG. The enzymatic reaction was halted by the addition of 2 M H_2_SO_4_ after introducing 200 μL of substrate solution (A: H_2_O_2_, B: 3,3′,5,5′-Tetramethylbenzidine). Each plate was equipped with standard controls (negative and positive). Ultimately, the results were quantified by measuring the absorbance at 450 nm using a microplate enzyme-linked immunosorbent assay reader (Thermo Scientific, San Jose, CA, USA). Serum IgE levels were determined using a goat IgE ELISA kit in accordance with the manufacturer’s instructions.

### 2.9. Determination of Cytokine Concentration

The indirect enzyme-linked immunosorbent assay (ELISA) was employed to quantify the serum concentrations of Interleukin (IL)-4, IL-9, IL-17, interferon-γ (IFN-γ), and transforming growth factor-β (TGF-β) at various time points, specifically on days 0, 14, 28, 43, 54, and 63 throughout the experimental duration. In this analysis, goat cytokine ELISA kits (IL-4 ml9024598, IL-9 ml60807530, IL-17 ml90156742, TGF-β ml20153590, and IFN-γ ml4767261) were utilized, all of which were procured from HengYuan, Shanghai, China PR. All procedural steps were meticulously carried out following the precise instructions provided by the kit manufacturer.

### 2.10. Parasitological Techniques

Fresh fecal samples were aseptically obtained via rectal collection from each goat at specific time points, namely, on days 50, 52, 54, 56, 58, 60, and 62 throughout the experimental timeline. Subsequently, the fecal egg count, also known as EPG (eggs per gram), was quantified by employing the modified McMaster method [31]. The reduction in EPG was calculated employing the following formula:RE (%) = (Positive group − Experimental group)/Positive group × 100%

Here, ‘RE’ denotes the reduction rate of the fecal egg count.

### 2.11. Abomasal Worm Loads

On the 63rd day of the experiment, all animals were humanely euthanized, and a thorough examination of the abomasum was conducted to determine the quantity of worms, including both male and female specimens [32]. The contents of the abomasum were systematically retrieved, and the mucosa was gently scratched and washed with a lukewarm 0.9% sodium chloride solution to dislodge the adhering maggots. Then, all worms from the abomasum and mucosal surface were collected carefully, enumerated, and classified on the basis of gender. The evaluation of declines in the total worm population was conducted by following the same approach employed for determining EPG.

### 2.12. Differential Cell Counts

To perform a differential blood cell count, blood samples were aseptically collected from all goats through jugular vein puncture. These samples were deposited into evacuated glass tubes (Becton Dickinson, Oxford, England) coated with ethylenediamine tetraacetic acid (EDTA) on specified days, including days 0, 14, 28, 43, and 63 of the experimental periods. The classification of blood cell types was accomplished using an automated electronic cell counter (Mindray BC-5000 Vet, Shenzhen, China).

### 2.13. Statistical Analysis

The data from all experiments were presented in the format of mean ± standard deviation (SD). To assess and clarify significant differences among the groups, the two-way ANOVA test with Tukey’s multiple comparisons test was employed. Significance levels for data differences were established as *p* < 0.05, *p* < 0.01, and *p* < 0.001 [17].

## 3. Results

### 3.1. The Working Concentration of PVA

The assessment of particle morphology and size was conducted using SEM at varying PVA concentrations, specifically 1%, 4%, and 6%. The particle size displayed irregularity at PVA concentrations of 1% and 4% and yielded NPs of more than 2 µm (Appendix A). Hence, 6% PVA was selected as the most appropriate for subsequent experiments.

### 3.2. Characteristics of Antigen-Loaded PLGA NPs

#### 3.2.1. Scanning Electron Microscopy of Antigen Encapsulating NPs

The SEM analysis demonstrated the smooth surface characteristics of the nanoparticles, with the size of rHcARF1-PLGA NPs ranging between 63 nm and 125 nm (Figure 1A,B).

#### 3.2.2. Analysis of SDS-PAGE

A 12% SDS-PAGE gel was utilized to evaluate the binding and integrity of rHcARF1 with PLGA NPs. The results unambiguously demonstrated that the molecular weight of the protein remained unaffected by the NP formulation. Specifically, the rHcARF1-PLGA NPs exhibited a prominent band with a size of approximately 38 kDa, while PBS-PLGA NPs (blank NPs) did not exhibit any conforming sign or band (Figure 1C). The western blot analysis validated the specific detection of the recombinant HcARF1 protein by the immune sera, notably the rat anti-rHcARF1, which manifested as a distinct band. In contrast, sera from normal rats exhibited no significant recognition of the HcARF1 protein, suggesting a lack of specific binding (Appendix A). While, densitometry readings of immunoblot are mentioned in Appendix A.

#### 3.2.3. Encapsulation Efficiency and Loading Capacity Assessment

Subsequent to the conjugation process, PLGA NPs were subjected to precipitation, and the residual amount of unbound rHcARF1 protein was quantified. The analysis demonstrated an encapsulation efficiency (EE) of 72.37 ± 3.51%, indicating the successful encapsulation of this proportion of rHcARF1 within the PLGA NPs. Furthermore, approximately 25 ± 1.1% of the protein was loaded (LC) by the PLGA NPs, as detailed in Table 2.

### 3.3. Nanovaccine Modulated the Sera IgG, IgA, and IgE

In the experimental group (rHcARF1-PLGA), a noteworthy increase in specific serum IgA levels was observed from the period of immunization to the challenge, reaching peak values on days 28 and 54 compared to positive and negative controls. Furthermore, serum IgG levels, particularly on day 43, showed a significant elevation in the nanovaccine group (rHcARF1-PLGA) compared to the positive and negative control groups (*p* < 0.05, *p* < 0.01, *p* < 0.001; Figure 2A,B).

A similar trend was evident in the total IgE serum levels of goats subjected to the same treatment as described earlier. The IgE serum levels in the experimental group exhibited a significant increase on day 28. They reached their zenith on day 43 (*p* < 0.01, *p* < 0.001), persisting at elevated levels until the conclusion of the study, in contrast to the positive and negative control groups (Figure 2C).

### 3.4. Nanovaccine Augmented the Production of Sera Cytokines

Subsequent to immunization with rHcARF1-PLGA, a conspicuous upsurge was detected in the production of cytokines, specifically IL-4 (days 14 to 63), IL-9 (days 28 to 54), IL-17 (days 54, 63), and TGF-β (days 54, 63) in all vaccinated animals. All these cytokines were increased and showed statistical significance (*p* < 0.05, *p* < 0.01, *p* < 0.001) when compared to the negative control group following the challenge (Figure 3). In the case of the positive control, IL-4 (days 43 to 63), IL-9 (days 43, 54), and IL-17 (days 63) concentrations were higher as compared to the negative control (*p* < 0.05, *p* < 0.01, *p* < 0.001). It is remarkable that goats immunized with the nanovaccine produced more IL-4 (days 43) and IL-9 (days 43) after the L3 challenge as compared with the positive control (*p* < 0.05). However, no substantial alterations were observed in the concentration of IFN-γ in any of the groups, both before and after the challenge with L3 larvae.

### 3.5. Nanovaccine Reduced the EPG and Worm Load

Figure 4 presents a comparison of EPG counts between vaccinated and unvaccinated goats after the challenge. Fecal egg count results demonstrated that egg shedding commenced in the experimental (rHcARF1-PLGA) and positive control groups on day 50 of the experiment. After day 54, this decrease was significant, and the trend continued until day 58. Subsequently, egg shedding in the experimental group decreased, and this reduction continued after day 62, resulting in a significant 47.5% decrease in comparison to the positive control group (*p* < 0.01, *p* < 0.001). The negative control group did not exhibit any egg shedding throughout the experimental period as it was not challenged with *H. contortus* L3.

Additionally, goats that received the nanovaccine displayed a noteworthy reduction in worm burdens by 55.7% when compared with unvaccinated goats (Figure 5, *p* < 0.001). This reduction was evident for both female (53.6%, *p* < 0.01) and male worms (59.7%, *p* < 0.05), underscoring the substantial efficacy of the nanovaccine against this helminth.

### 3.6. Nanovaccine Modulated the Blood Cell Counts

The peripheral basophil counts displayed a significant increase after immunization (days 43 and 63, *p* < 0.05, *p* < 0.001) in the vaccinated group when compared to the positive and negative control groups (Figure 6A). Notably, a significant distinction was observed between the experimental and positive groups. Subsequent to the second immunization, the eosinophil count in vaccinated animals exhibited a remarkable elevation (*p* < 0.05) from days 28 to 63 in contrast to the negative group (Figure 6B). However, the goats that received nanovaccine did not show a remarkable difference as compared to the positive control group in this case.

According to Figure 6C, continuous elevation in blood lymphocyte counts was noted in the immunized group after the secondary immunization, with a significant difference compared to the negative control group observed from days 43 to 63 (*p* < 0.001). The increasing trend of lymphocytes was obvious as compared to positive control also on days 63 (*p* < 0.05).

Concerning hemoglobin concentration, a non-significant reduction was observed in the rHcARF1-PLGA and positive control groups after the challenge with L3 larvae (days 28–63) in comparison to the worm-free goats in the negative control group (Figure 6D). Moreover, no significant differences were detected in the hematocrit values for monocytes and neutrophils among all groups during the experiment (Figure 6E,F).

## 4. Discussion

Vaccination strategies for combating *Haemonchus contortus* (*H. contortus*) infections are of paramount importance in the realm of livestock research. The intricate and dynamic life cycle of *H. contortus* has thus far posed a formidable challenge in the development of an effective vaccine that can boost natural immunity. Notably, the parasite’s larvae undergo antigenic and molecular transformations with each molting stage [33,34], underscoring the necessity of identifying protective candidate antigens tailored to the specific parasite stage. In this study, we assessed the protective immunity elicited by rHcARF1, in conjunction with the nanomaterial PLGA, against *H. contortus* infection in goats. The vaccine formulation consisted of the helminth recombinant protein rHcARF1, efficiently encapsulated within the biodegradable polymer PLGA NPs. This choice was made following previous findings confirming the capacity of this antigenic molecule to modulate the cellular and humoral immune response of host immune cells [15,16].

The integration of recombinant proteins and nanospheres in vaccines against *H. contortus* presents notable scientific advantages. Recombinant proteins are produced with high specificity and purity, utilizing control systems that ensure only the targeted antigens are included. This precision minimizes contamination risks and allows for the incorporation of specific epitopes that can trigger a strong immune response [17,27]. Additionally, the consistent production of recombinant proteins ensures uniformity in vaccine quality and effectiveness while eliminating the risks associated with vaccines derived from live or killed pathogens [35]. PLGA-based nanospheres enhance vaccine performance by providing controlled, sustained release of antigens and targeted delivery to antigen-presenting cells. This approach improves the durability of the immune response and stabilizes the antigen, minimizing side effects [36,37,38]. Together with recombinant proteins, these nanospheres optimize the vaccine’s potency, stability, and safety against *H. contortus*.

Numerous experimental studies have previously demonstrated the efficacy of DNA vaccines in reduced egg shedding and diminishing worm loads in the context of this helminth infestation (*H. contortus*) [27,32,39]. In our present investigation, we consistently observed reductions in both EPG counts and abomasal worm burden. This achievement represents a noteworthy exemplar of a nanoparticle-based approach that has yielded a modest yet promising response. It introduces an innovative concept of combining potential helminth antigens with a biopolymer, specifically PLGA. Nevertheless, it is important to acknowledge that EPG can exhibit variability based on factors such as the immune status of the animals, their age, and the specific strains of *H. contortus* [40].

Antibodies are widely recognized as critical components in providing significant immunity against *H. contortus* [41]. Particularly, antibodies of the IgG, IgA, and IgE classes bind to antigens, forming immune complexes that activate an inflammatory cascade. This response leads to increased mucus production, smooth muscle contractions, and, ultimately, the expulsion or death of the parasite [42,43]. Goats in the experimental group in the current investigation exhibited a significant increase in IgG titers, particularly after booster vaccinations, compared to the unvaccinated control group. These findings are consistent with our previous research, where the Dim-1 protein of *H. contortus* induced significantly elevated IgG levels compared to control groups [32]. Similar responses have also been documented in other nematode infections, including *Ascaris suum* [44] and *Trichostrongylus colubriformis* [45]. Collectively, these results underscore the ability of rHcARF1 to elicit a robust immune response in the host.

In our investigation, we observed a significant elevation (*p* < 0.05, *p* < 0.01) in serum-specific IgA levels in response to the nanovaccine on days 28 and 54, as compared to the positive control group. These findings align with previous research indicating a close relationship between abomasal IgA levels and worm burdens associated with various parasites [46,47,48]. Among the different immunoglobulin isotypes, IgE has consistently demonstrated associations with nematode infections [43,49,50]. Furthermore, studies involving resistant sheep have revealed elevated production levels of IgG1 and IgE [51]. In our current study, the results revealed a significant increase in total serum IgE levels in the experimental groups at days 28 and 43 (*p* < 0.01, *p* < 0.001) when compared to the negative control group. These collective findings underscore the potential importance of IgG, IgA, and IgE in inducing protective immunity against *H. contortus*.

Regarding immune responses to helminth infections, eosinophils play a crucial role in providing resistance to parasites in ovine species [52]. Eosinophilia, characterized by an increase in eosinophils, has been observed in various allergic diseases as well as parasitic infections [53]. In our current investigation, we observed elevated levels of eosinophils, basophils, and lymphocytes in the blood of the immunized group in comparison to the negative control group. This data is consistent with the common knowledge of the strong connection between eosinophils and the ability to resist helminth parasites [54]. It is well-established that *H. contortus* infection often leads to a reduced hemoglobin level. A previous study also reported lowered hemoglobin levels in the blood of vaccinated animals [32]. Consequently, the low hemoglobin level suggests that hemoglobin values may not serve as a reliable indicator of worm burdens in infected animals. We found a miniature increase in the hemoglobin level of the nanovaccine group on day 14 before the L3 challenge, which probably means that the antigen (rHcARF1) started to produce its immune response. However, after the L3 challenge, the protective effect was diminished, as shown by the hemoglobin level. In our present study, we did not observe significant differences in the values of two blood fractions (monocytes, neutrophils) between the experimental and positive groups. It is conceivable that the infection of *H. contortus* L3 larvae contributed to the increased values in both groups. However, further investigations are required to provide a more comprehensive explanation.

IL-4 is the quintessential cytokine signaling the Th2 immune response and is chiefly responsible for triggering the switch to the IgE isotype [55]. Elevated IL-4 levels in *H. contortus* infections have been linked to an increased production of IgE directed against *H. contortus* [56]. Notably, cells isolated from the abomasal and mesenteric lymph nodes of *H. contortus*-infected lambs exhibited reduced IFN-γ expression compared to uninfected counterparts [51]. TGF-β, a powerful regulator, exerts dual roles by suppressing both Th1 and Th2 cells while fostering the maintenance and function of regulatory T (Treg) cells. Additionally, TGF-β has significant implications in hematopoiesis and plays pivotal roles in embryogenesis, tissue regeneration, cell proliferation, and differentiation [57]. A body of experimental evidence underscores the essential role of IL-17 as a driving force behind the recruitment and activation of neutrophils [58]. Furthermore, IL-9, a well-established Th2-associated cytokine, amplifies the biological function of IL-4 in expediting the expulsion of worms [59]. Our study unveiled that the concentration of IL-4 in vaccinated animals (from day 14 to day 63) significantly surpassed that in the negative control group. The current research recorded an elevated production of IL-9 (from days 28 to 54), IL-17 (from days 54 to 63), and TGF-β (from days 54 to 63) in the nanovaccine group relative to the control group (*p* < 0.05, *p* < 0.01, *p* < 0.001). Furthermore, the goats that received the nanovaccine secreted more IL-4, IL-9, IL-17, and TGF-β on days 14, 28, 43, 54, and 63 compared to the positive control, respectively. Simultaneously, we observed no significant differences in IFN-γ concentration across all groups. In light of our understanding of cytokines, it is plausible to hypothesize that rHcARF1 plays a critical role in the pathogenesis and elicitation of inflammatory responses in *H. contortus* infections.

The rHcARF1-PLGA nanovaccine demonstrated partial protection against *H. contortus* infection in goats; however, several important limitations must be addressed. First, although reductions in eggs per gram (EPG) of feces (47.5%) and worm burden (55.7%) were statistically significant, the level of protection was incomplete. This indicates a need for further refinement of the antigen dose, formulation, or vaccination schedule to enhance efficacy. Second, the relatively small sample size (n = 15) limits the generalization of these findings, necessitating larger, more comprehensive trials to validate the protective effects of the vaccine and assess its applicability in broader contexts. Lastly, this study did not include a direct comparison with the licensed vaccine (Barbervax^®^) for Haemochosis. Future investigations should focus on such comparisons to determine the relative efficacy, long-term immune response, and cost-effectiveness of the rHcARF1-PLGA nanovaccine in relation to existing immunological products.

## 5. Conclusions

Our research shows that the nanovaccine effectively protects goats from *H. contortus* infestations. It significantly reduces egg shedding and worm loads in the abomasum. The vaccine also induces strong immune responses, marked by increased cytokine levels in vaccinated goats. By combining the antigen with PLGA nanoparticles, this approach enhances antibody production and improves the host’s defense against *H. contortus*. However, more detailed studies are needed to elucidate the complex mechanisms through which HcARF1 contributes to this protective effect.

## Figures and Tables

**Figure 1 vaccines-12-01188-f001:**
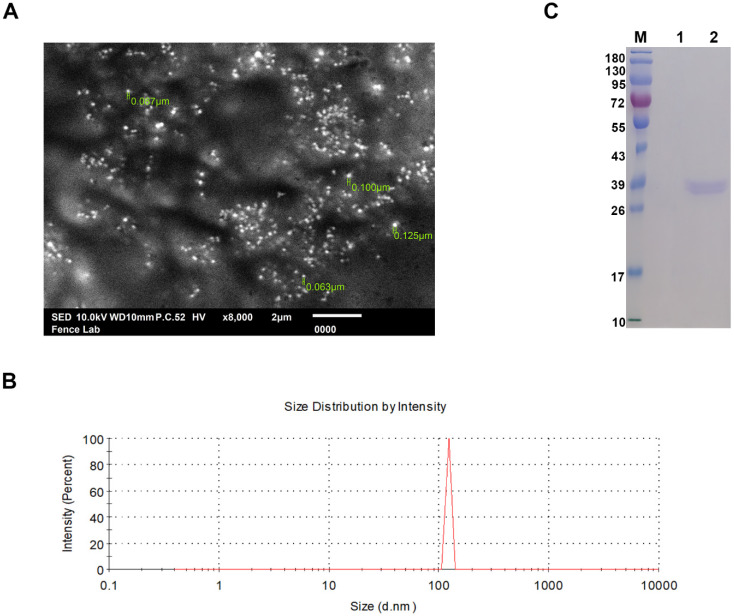
The scanning electron microscope was employed to ascertain the morphology and size characteristics of nanoparticles (NPs). Their morphology was visualized at a magnification of 10,000×. In (**A**), the SEM image depicts rHcARF1-PLGA NPs, (**B**) provides a graphical representation of the size distribution of rHcARF1-PLGA NPs, and (**C**) showcases an SDS-PAGE analysis with a 12% separating gel, conducted to investigate the interaction between rHcARF1 and PLGA nanoparticles. In this analysis, Lane M represents the standard protein molecular weight marker; Lane 1 indicates PBS-PLGA NPs and Lane 2 demonstrates PLGA NPs bound with rHcARF1.

**Figure 2 vaccines-12-01188-f002:**
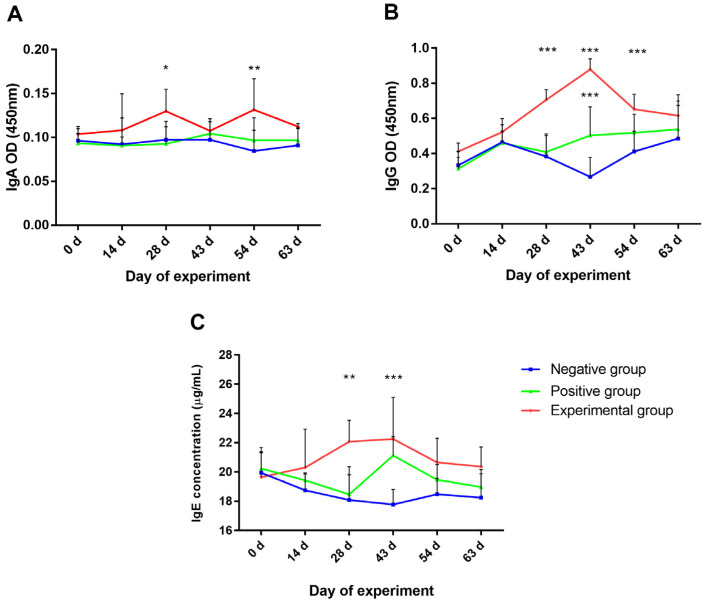
The quantification of rHcARF-1 sera immunoglobulin levels (IgG, IgA, and IgE) was carried out using both spectrophotometry and enzyme-linked immunosorbent assay (ELISA) techniques. In (**A**–**C**), the IgA, IgG and IgE titers of the rHcARF-1 experimental group, the negative control group, and the positive control group were reported as mean values with standard deviation (mean ± SD) based on optical density measurements taken at 450 nm. The presented data are representative of triplicate experiments, and statistical significance levels are indicated as follows: * *p* < 0.05, ** *p* < 0.01, and *** *p* < 0.001.

**Figure 3 vaccines-12-01188-f003:**
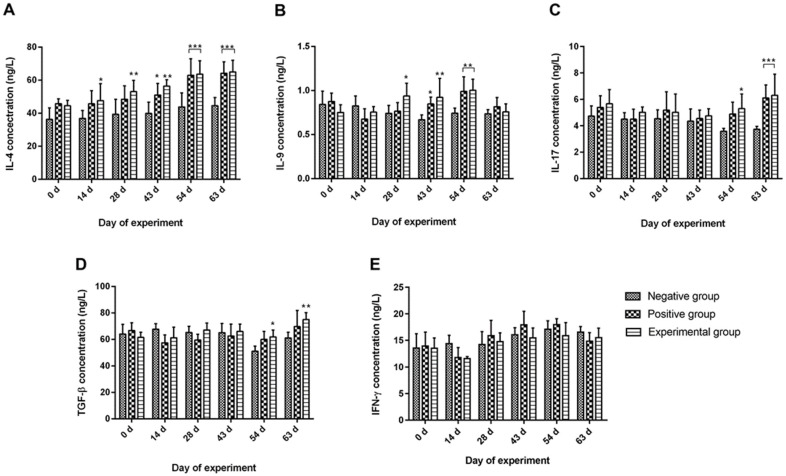
Sera cytokine levels, including IL-4, IL-9, IL-17, TGF-β, and IFN-γ, were quantified and are expressed as mean values accompanied by standard deviation (mean ± SD). The concentration of IL-4, (**A**), IL-9 (**B**), IL-17 (**C**), TGF-β (**D**), and IFN-γ (**E**). The experimental animals received immunization on day 0 and day 14, followed by a challenge with L3 on day 28. The presented data are representative of triplicate experiments, and statistical significance levels are indicated as follows: * *p* < 0.05, ** *p* < 0.01 and *** *p* < 0.001.

**Figure 4 vaccines-12-01188-f004:**
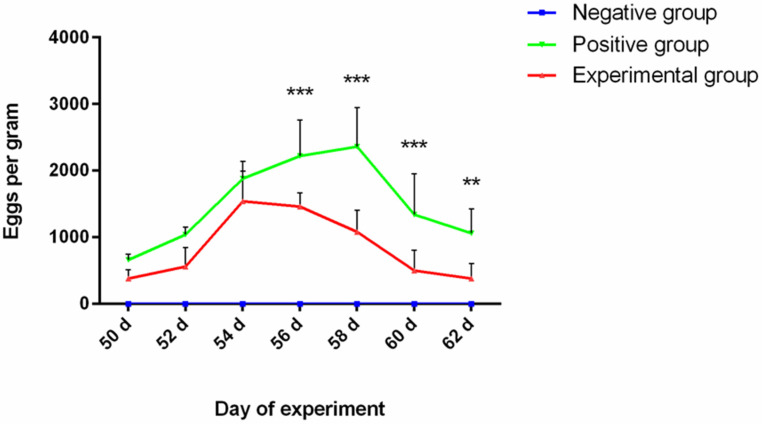
The reduction in egg shedding, quantified as eggs per gram (EPG), was expressed as the mean value along with its standard deviation (mean ± SD). The experimental groups included goats that received vaccinations with rHcARF1-PLGA and were subsequently challenged, designated as the rHcARF1-PLGA group. The positive control group comprised goats injected with PBS only and subjected to the same challenge, while the negative control group consisted of uninfected and unimmunized goats. The presented data are representative of triplicate experiments, and statistical significance levels are indicated as follows: ** *p* < 0.01 and *** *p* < 0.001.

**Figure 5 vaccines-12-01188-f005:**
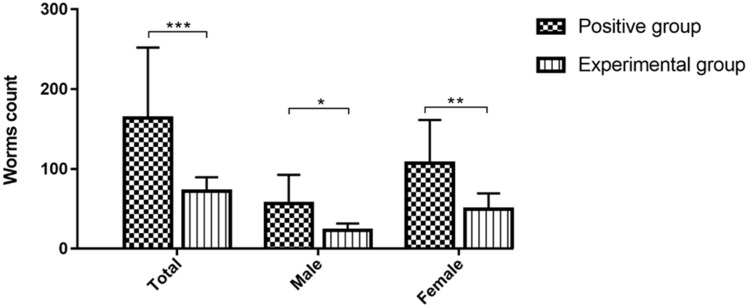
Quantification of female, male, and total worm counts was conducted across multiple experimental groups. The outcomes are reported as the mean values accompanied by their standard error of the mean (mean ± SEM). The experimental cohort receiving the nanovaccine and subsequent challenge is designated as the rHcARF1-PLGA group. The positive control group comprises goats exclusively administered with PBS and subsequently challenged, while the negative control group encompasses goats that remained uninfected and devoid of immunization. The presented data are a representative of statistical significance at levels * *p* < 0.05, ** *p* < 0.01 and *** *p* < 0.001.

**Figure 6 vaccines-12-01188-f006:**
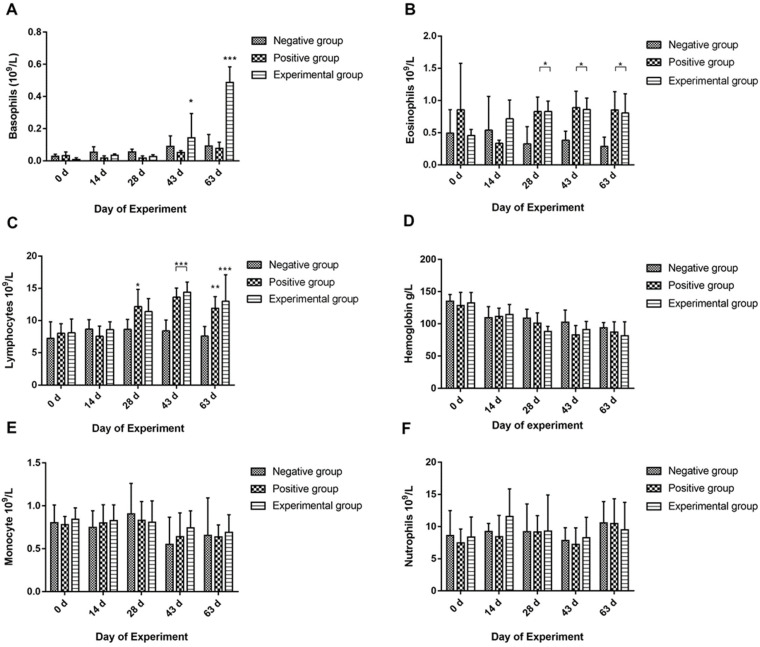
The quantification of basophils, eosinophils, neutrophils, lymphocytes, monocytes, and hemoglobin levels in the blood is presented as the mean values accompanied by their respective standard errors of the mean (means ± SEM). The fluctuation of different blood fractions, such as basophils (**A**), eosinophils (**B**), lymphocytes (**C**), hemoglobin (**D**), monocytes (**E**) and neutrophils (**F**) are shown in this figure. Within the study, the rHcARF1-PLGA group signifies goats that were vaccinated with the nanovaccine and subsequently challenged, whereas the positive control group pertains to goats exclusively administered with PBS and subsequently subjected to the same challenge. The negative control group encompasses goats that remained uninfected and were not subjected to any immunization. Immunization of the animals occurred on days 0 and 14, followed by a challenge involving 8000 *H. contortus* L3 larvae on day 28. The presented data are a representative of triplicate experiments, denoted as statistical significance at levels * *p* < 0.05, ** *p* < 0.01 and *** *p* < 0.001.

**Table 1 vaccines-12-01188-t001:** Schedule for vaccinations, challenge with infection, and sampling.

Day of Experiment	0	14	28	43	50	52	54	56	58	60	62	63
Vaccination/Immunization	*	*										
Challenge with L3			*									
Serum collection for IgG, IgA, and IgE	*	*	*	*			*					*
Feces collection for fecal egg count					*	*	*	*	*	*	*	
Worms count												*

‘*’ means the occurance of event in the study perios.

**Table 2 vaccines-12-01188-t002:** Characterization of recombinant antigen (rHcARF1) loaded-PLGA NPs. Data are presented as the mean ± SD (n = 3).

NPs	Size (nm)	LC ^a^	EE ^b^
rHcARF1-PLGA NPs	100 ± 36	25 ± 1.1	72.37 ± 3.51

LC ^a^ = (total protein − unbound protein)/total dry weight of Nano-vaccine × 100%. EE ^b^ = (total protein − unbound protein)/total protein × 100%.

## Data Availability

The datasets supporting the conclusions of this article are included in the article.

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
