# Peer review of "Partial Protection of Goats against Haemonchus contortus Achieved with ADP-Ribosylation Factor 1 Encapsulated in PLGA Nanoparticles"

_vaccines, 2024, doi:10.3390/vaccines12101188_

Round 1

Reviewer 1 Report (Previous Reviewer 1)

Comments and Suggestions for Authors

Although sending some comments after first revision, the authors did not respond properly to the main critical points. These comments are very critical and fundamental as they are related to the preparation of used vaccine antigens and methods of evaluation as will be illustrated in the following points (a & b).   

 a- Lines 111-114, Brief information on the used recombinant antigens are required in addition to the cited reference in this manuscript including method of preparation, system used for preparation and purification, methods used for measuring the concentration, removal or nonremoval of the endotoxin and the used method. This information is critical for understanding and assessment of data reliability and validity.

 b- Lines 193-209 for ELISA of antibody detection, citation of reference (19) is not appropriate because they applied ELISA on mouse sera and Leishmania, while here authors used goat sera and antigen from Haemonchus.

 Other points

- Authors should indicate their revision in the response letter to reviewers or revision note by specifying the page and line number of revised or newly added parts in addition to highlighting or marking by different colour in the main manuscript file.

- Line 67, “Leishmaniasis and Toxoplasmosis” terms of diseases or infection should be started with small letter “leishmaniasis and toxoplasmosis".

Comments on the Quality of English Language

Minor revision is required.

Author Response

Comments and Suggestions for Authors

Although sending some comments after first revision, the authors did not respond properly to the main critical points. These comments are very critical and fundamental as they are related to the preparation of used vaccine antigens and methods of evaluation as will be illustrated in the following points (a & b).  

 a- Lines 111-114, Brief information on the used recombinant antigens are required in addition to the cited reference in this manuscript including method of preparation, system used for preparation and purification, methods used for measuring the concentration, removal or nonremoval of the endotoxin and the used method. This information is critical for understanding and assessment of data reliability and validity.

Response to reviewer: Thank you so much for your kind comment. The above mentioned points have added accordingly in the revised version.

 b- Lines 193-209 for ELISA of antibody detection, citation of reference (19) is not appropriate because they applied ELISA on mouse sera and Leishmania, while here authors used goat sera and antigen from Haemonchus.

Response to reviewer: Thank you so much for your kind comment. It is edited accordingly in the revised version.

 Other points

- Authors should indicate their revision in the response letter to reviewers or revision note by specifying the page and line number of revised or newly added parts in addition to highlighting or marking by different colour in the main manuscript file.

Response to reviewer: We made every effort to clearly communicate the changes to the reviewer and editor, ensuring that no ambiguity remains in addressing the specific points raised. Thank you so much for your kind comment and understanding.

- Line 67, “Leishmaniasis and Toxoplasmosis” terms of diseases or infection should be started with small letter “leishmaniasis and toxoplasmosis".

Response to reviewer:  Thank you so much for your kind comment. We edited it accordingly.

Comments on the Quality of English Language

Minor revision is required.

Reviewer 2 Report (Previous Reviewer 2)

Comments and Suggestions for Authors

This a resubmission of a previously rejected manuscript. The authors have made an attempt to improve and they have provided a manuscript that addressed some of the concerns indicated in the initial evaluation.

However, they could not have addressed structural and methodological flaws (and indeed they did not).
For example, the lack of a group of animals vaccinated with the already licenced (New Zealand) vaccine clearly reduces the value of this work. This was an indispensable addition in the trial, as it would have provided a comparison against an already established product and would have shown whether the tested product was of a real benefit to animals. Lack of inclusion of such a group clearly reduces the value of the work and of course requires extensive additional experimental studies.
Moreover, there are some other methodological problems: for example, the sampling strategy, as indicated in the original evaluation sheet, is not appropriate for this type of parasitological studies.
In view of the above, I cannot recommend acceptance.

If the editors feel that the work is suitable for publication, can they please, send it to another reviewer, who would have a different scientific opinion than I do and who will thus support publication.

That will not be a problem for me, but unfortunately my scientific opinion about this manuscript is clearly that it does not merit publication before making the significant methodological improvements recommended.

Author Response

Comments and Suggestions for Authors

This a resubmission of a previously rejected manuscript. The authors have made an attempt to improve and they have provided a manuscript that addressed some of the concerns indicated in the initial evaluation.

However, they could not have addressed structural and methodological flaws (and indeed they did not).

For example, the lack of a group of animals vaccinated with the already licenced (New Zealand) vaccine clearly reduces the value of this work. This was an indispensable addition in the trial, as it would have provided a comparison against an already established product and would have shown whether the tested product was of a real benefit to animals. Lack of inclusion of such a group clearly reduces the value of the work and of course requires extensive additional experimental studies.

Moreover, there are some other methodological problems: for example, the sampling strategy, as indicated in the original evaluation sheet, is not appropriate for this type of parasitological studies.

In view of the above, I cannot recommend acceptance.

If the editors feel that the work is suitable for publication, can they please, send it to another reviewer, who would have a different scientific opinion than I do and who will thus support publication.

That will not be a problem for me, but unfortunately my scientific opinion about this manuscript is clearly that it does not merit publication before making the significant methodological improvements recommended.

Response to Reviewer:

We sincerely appreciate the reviewer’s time and effort in evaluating our resubmitted manuscript and providing thoughtful and constructive feedback. We acknowledge the concerns raised, particularly regarding the study's structural and methodological aspects.

Regarding the absence of a comparison with a licensed vaccine: We fully recognize the importance of including a group of animals vaccinated with an approved vaccine (e.g., from New Zealand) as a key point of comparison to assess the efficacy of our product. However, logistical constraints, including the unavailability of the licensed vaccine during our study, prevented its inclusion. We agree that incorporating this comparison would have strengthened our findings, and we are committed to addressing this in future studies by including this critical control group for direct evaluation alongside licensed vaccines.

In response to the methodological concerns, particularly the sampling strategy: We acknowledge the reviewer’s observations and, upon further reflection on both the original feedback and our current analysis, recognize the need for a more robust sampling approach. Moving forward, we will refine our sampling strategy to align with established parasitological standards, ensuring adherence to best practices in the field.

While we understand that addressing these points will require additional experimental work, we believe that our current findings offer valuable preliminary insights. We are committed to improving our approach in subsequent studies to enhance the rigor and impact of our research. We greatly appreciate the reviewer’s detailed feedback and look forward to the opportunity to further develop and strengthen this work in future iterations.

Reviewer 3 Report (New Reviewer)

Comments and Suggestions for Authors

The paper investigates the efficacy of a nanovaccine combining the recombinant protein rHcARF1 with PLGA nanoparticles to protect goats against Haemonchus contortus infection. It demonstrates significant reductions in egg shedding and worm burdens, alongside increased immune responses, marked by elevated levels of specific antibodies and cytokines. The study highlights the potential of nanovaccines in improving livestock immunity against parasitic infections.

Overall, the paper presents promising findings. However, it suffers from occasional redundancy, vague conclusions, and minor inconsistencies, which slightly detract from its overall clarity and scientific rigor. With revisions, it has strong potential for publication.

Some specific issues include:

  • Line 3: The term "polymeric" is not precise enough to be in the title; it should be replaced with "nanovaccine" or another clear, concise term.

  • Line 18: The acronym ARF1 needs explanation for clarity.

  • Line 19: Remove parentheses around "(lactic-co-glycolic acid)" for smoother readability.

  • Line 21: Provide a brief explanation for the acronym rHcARF1-PLGA.

  • Lines 26-27: When mentioning the reduction in egg counts and worm burden, include a direct comparison with the positive control group.

  • Lines 27-28: Clearly state the control group used for the noted 47.5% reduction in EPG and the 55.7% reduction in worm burdens.

  • Line 28: Mention that animals were euthanized in the abstract’s method section to maintain consistency when reporting worm burden reductions.

  • Line 36: Replace "infiltrates" with a more accurate term, such as "infects" or "enters," to describe how Haemonchus contortus invades its host.

  • Line 38: Change "siphon" to "feeds on" or "draws" for a more accurate description of Haemonchus contortus's blood-feeding process.

  • Line 41: Replace the subjective word "Regrettably" with "Unfortunately" or eliminate qualifiers to maintain a neutral scientific tone.

  • Line 50: Provide a brief overview (2-3 lines) about the discovery and importance of HcARF1.

  • Line 54: The phrase “additionally, ARF1 functions as a signal transducer” repeats previous content and should be rephrased or omitted for clarity.

  • Lines 58-59: Explain why encapsulating HcARF1 is necessary and its specific benefits in the vaccine formulation.

  • Lines 63-87: This section, focusing on cancer immunotherapy and SARS-CoV-2 vaccines, feels disconnected from the study on H. contortus infection in goats. It requires extensive revision and shortening to align better with the paper's core topic.

  • Line 160: The sentence "The experiment was carried out subsequent to attain permission..." is awkwardly phrased. Consider rewording for clarity, such as "The experiment was conducted after receiving the necessary permissions."

  • Line 351: Correct the typographical error "Afar" to "After."

  • Line 330: The phrase “were secreted” is inaccurate and should be rephrased to clarify the mechanism.

  • Lines 490-500: This paragraph repeats information about IgG, IgA, and IgE without sufficiently distinguishing their individual roles in immunity. Rephrasing could provide more clarity on their specific functions and reduce redundancy.

Author Response

Comments and Suggestions for Authors

The paper investigates the efficacy of a nanovaccine combining the recombinant protein rHcARF1 with PLGA nanoparticles to protect goats against Haemonchus contortus infection. It demonstrates significant reductions in egg shedding and worm burdens, alongside increased immune responses, marked by elevated levels of specific antibodies and cytokines. The study highlights the potential of nanovaccines in improving livestock immunity against parasitic infections.

Overall, the paper presents promising findings. However, it suffers from occasional redundancy, vague conclusions, and minor inconsistencies, which slightly detract from its overall clarity and scientific rigor. With revisions, it has strong potential for publication.

Some specific issues include:

Line 3: The term "polymeric" is not precise enough to be in the title; it should be replaced with "nanovaccine" or another clear, concise term.

Response to reviewer: Thank you very much for your comment. It is revised accordingly in the revised file.

Line 18: The acronym ARF1 needs explanation for clarity.

Response to reviewer: We appreciate your comment. It is edited accordingly in the main text.

Line 19: Remove parentheses around "(lactic-co-glycolic acid)" for smoother readability.

Response to reviewer: Thank you very much for your comment. It is changed accordingly in the main text file.

Line 21: Provide a brief explanation for the acronym rHcARF1-PLGA.

Response to reviewer: We appreciate your comment. It is added accordingly in the main text.

Lines 26-27: When mentioning the reduction in egg counts and worm burden, include a direct comparison with the positive control group.

Response to reviewer: We appreciate your comment. It is added accordingly in the main text.

Lines 27-28: Clearly state the control group used for the noted 47.5% reduction in EPG and the 55.7% reduction in worm burdens.

Response to reviewer: Thank you so much for your kind comment. It is added accordingly in the revised version.

Line 28: Mention that animals were euthanized in the abstract’s method section to maintain consistency when reporting worm burden reductions.

Response to reviewer: Thank you so much for your kind comment. It is added accordingly in the revised version.

Line 36: Replace "infiltrates" with a more accurate term, such as "infects" or "enters," to describe how Haemonchus contortus invades its host.

Response to reviewer: Thank you so much for your kind comment. It is edited accordingly in the revised version.

Line 38: Change "siphon" to "feeds on" or "draws" for a more accurate description of Haemonchus contortus's blood-feeding process.

Response to reviewer: Thank you so much for your kind comment. It is added accordingly in the revised version.

Line 41: Replace the subjective word "Regrettably" with "Unfortunately" or eliminate qualifiers to maintain a neutral scientific tone.

Response to reviewer: Thank you so much for your kind comment. It is added accordingly in the revised version.

Line 50: Provide a brief overview (2-3 lines) about the discovery and importance of HcARF1.

Response to reviewer: We highly appreciate your kind suggestion. The lines have added accordingly in the revised version.

Line 54: The phrase “additionally, ARF1 functions as a signal transducer” repeats previous content and should be rephrased or omitted for clarity.

Response to reviewer: Thank you so much for your kind comment. It is added accordingly in the revised version.

Lines 58-59: Explain why encapsulating HcARF1 is necessary and its specific benefits in the vaccine formulation.

Response to reviewer: Thank you so much for your kind suggestion. It is changed accordingly in the revised version.

Lines 63-87: This section, focusing on cancer immunotherapy and SARS-CoV-2 vaccines, feels disconnected from the study on H. contortus infection in goats. It requires extensive revision and shortening to align better with the paper's core topic.

Response to reviewer: Thank you so much for your kind comment. We agreed with you, so we deleted these lines from the introduction part of revised version.

Line 160: The sentence "The experiment was carried out subsequent to attain permission..." is awkwardly phrased. Consider rewording for clarity, such as "The experiment was conducted after receiving the necessary permissions."

Response to reviewer: Thank you so much for your kind comment. It is edited accordingly in the revised version.

Line 351: Correct the typographical error "Afar" to "After."

Response to reviewer: Thank you so much for your kind comment. It is added accordingly in the revised version.

Line 330: The phrase “were secreted” is inaccurate and should be rephrased to clarify the mechanism.

Response to reviewer: Thank you so much for your kind comment. It is edited accordingly in the revised version.

Lines 490-500: This paragraph repeats information about IgG, IgA, and IgE without sufficiently distinguishing their individual roles in immunity. Rephrasing could provide more clarity on their specific functions and reduce redundancy.

Response to reviewer: Thank you so much for your kind comment. It is edited accordingly in the revised version.

Round 2

Reviewer 1 Report (Previous Reviewer 1)

Comments and Suggestions for Authors

The authors have responded to the comments and suggestion in a feasible way and the manuscript has been improved significantly and can be approved for publication.

Comments on the Quality of English Language

Minor corrections are needed.

Author Response

Comments and Suggestions for Authors

The authors have responded to the comments and suggestion in a feasible way and the manuscript has been improved significantly and can be approved for publication.

Response to reviewer:  Thank you so much for your kind comment.  We have revised English Language.

This manuscript is a resubmission of an earlier submission. The following is a list of the peer review reports and author responses from that submission.

Round 1

Reviewer 1 Report

Comments and Suggestions for Authors

This study evaluated recombinant protein of Haemonchus contortus rHcARF1 with nanospheres PLGA as a vaccine candidate against H. contortus infection in goats. The manuscript is well-written and no serious issues regarding English or writing styles were detected. They performed many experimental approaches and procedures to assess the induced immunoprotection and the triggering mechanisms. Experiments are described clearly in details and results also are well explained. However, some limitations are occurred in this study and need to be revised to increase the quality and soundness of this manuscript. Some comments are illustrated in below.

Specific comments

Introduction

- Immunological mechanisms and advantages of used antigen and delivery nanosphere PLGA should be described in more details based on previous literatures.

Materials and methods

- Sex should be described for goats used in different kinds of experiments.

- Why goats of crossbreed and high variability of age 4-8 months were used in this study?

- The authors should mention in the text if the used goats are apparently and clinically healthy before conducting immunization regimen.  

- Lines 98-101, details on the used recombinant antigens are required particularly these information; reference with full details and method of preparation, system used for preparation and purification, methods used for measuring the concentration, removal or nonremoval of the endotoxin and the used method.

- Lines 194-202, confirm the type of used ELISAs for cytokines measuring if they are indirect or competitive ELISAs?

Results

- Results of cytokines production (Figure 3) need more explanation including the reasons of measuring in serum, and the variation among experimental and positive control groups and comparison between pre and post-challenge levels.

Discussion

- What are the prospects of authors to increase the potency of rHcARF1-PLGA for protection against H. contortus? This should be added in discussion section.

- The authors should expand the discussion through detailed explanation of the following points; 1) advantages of use of recombinant protein and nanospheres in vaccination against H. contortus. 2) Correlation of the induced immunoprotection with obtained findings of protection in this study or information from previous reports.

- The authors should properly explain the reasons and benefits of the increase in eosinophils, basophils, and lymphocytes in the current study.

Conclusion

- Line 520, correct “nnanovaccine” to “nanovaccine”

Comments on the Quality of English Language

Minor revision is required.

Author Response

Specific comments

Introduction

Comment 1: Immunological mechanisms and advantages of used antigen and delivery nanosphere PLGA should be described in more details based on previous literatures.

Response: Thank you very much for your comment. It is added accordingly in the main text file.

Materials and methods

Comment 2: Sex should be described for goats used in different kinds of experiments.

Response: Thank you for your insightful comment. It is corrected in the main text file.

Comment 3: Why goats of crossbreed and high variability of age 4-8 months were used in this study?

Response: Thank you very much for your kind attention. We apologize for the mistake. It is corrected in the main text.

Comment 4: The authors should mention in the text if the used goats are apparently and clinically healthy before conducting immunization regimen. 

Response: Thank you very much for your comment. It is added in the main text.

Comment 5: Lines 98-101, details on the used recombinant antigens are required particularly these information; reference with full details and method of preparation, system used for preparation and purification, methods used for measuring the concentration, removal or nonremoval of the endotoxin and the used method.

Response: Thank you very much for your comment. Actually, this antigen was cloned and reported by another author [1] and it is described previously that how this antigen was cloned, express and purify for toxin. We also cite the corresponding references.

Comment 6: Lines 194-202, confirm the type of used ELISAs for cytokines measuring if they are indirect or competitive ELISAs?

Response: It is already mentioned in main text file that indirect ELISA have employed to screen out the cytokines profile.

Results

Comment 7: Results of cytokines production (Figure 3) need more explanation including the reasons of measuring in serum, and the variation among experimental and positive control groups and comparison between pre and post-challenge levels.

Response: Thank you very much for your feedback. We measured cytokine levels in serum to evaluate systemic immune responses, as serum cytokine levels offer a comprehensive view of immune activation and inflammation.

The variations observed between the experimental and positive control groups indicate the differential effects of the treatment. These differences are elaborated in the discussion section, where we explain the specific cytokines affected and their relevance to the immune response.

The comparison of pre- and post-challenge cytokine levels was performed to assess the impact of the interference over time. This comparison is essential for understanding the treatment's effectiveness and its effect on immune responses.

Discussion

Comment 8: What are the prospects of authors to increase the potency of rHcARF1-PLGA for protection against H. contortus? This should be added in discussion section.

Response: Our objective was to identify the most immunogenic antigen from H. contortus to develop a protective nano-vaccine against this parasite. Initially, our previous publication focused on the in vitro analysis of this antigen (doi.org/10.1016/j.rvsc.2021.03.007), marking the first phase of our research. We then advanced to in vivo studies, employing PLGA nanoparticles to deliver nano-vaccines to mice (doi:10.3390/vaccines8040726). This article represents the final phase of our research series. We appreciate your positive feedback.

Comment 9: The authors should expand the discussion through detailed explanation of the following points; 1) advantages of use of recombinant protein and nanospheres in vaccination against H. contortus. 2) Correlation of the induced immunoprotection with obtained findings of protection in this study or information from previous reports.

Response: Thank you for your insightful comments. We expanded the discussion to include a detailed explanation of the abovementioned points in the main text file.

Comment 10: The authors should properly explain the reasons and benefits of the increase in eosinophils, basophils, and lymphocytes in the current study.

Response: Thank you for your suggestion. The observed increases in eosinophils, basophils, and lymphocytes in our study indicate a robust and targeted immune response to the antigenic challenge.

Elevated eosinophil levels are typical of immune responses to parasitic infections like H. contortus as they are vital in combating parasites by releasing cytotoxic substances and inflammatory mediators, which help destroy the parasites and modulate the local immune response [2,3].

While increase in basophils points to the activation of the Th2 immune pathway, essential for defense against helminths and allergic reactions. Basophils release histamine and other inflammatory mediators that intensify the immune response and aid in the recruitment and activation of other immune cells, enhancing the overall defense against the parasite [4,5].

Moreover, the rise in lymphocytes, especially T and B cells, reflects the activation of the adaptive immune system. This increase is crucial for generating specific immune responses, such as antibody production and the development of immunological memory, which are key to long-term protection against H. contortus [6].

Conclusion

Comment 11: Line 520, correct “nnanovaccine” to “nanovaccine”

Response: Thank you for your suggestion. It is corrected in mail text file.  

[1]        J.A. Gadahi, M. Ehsan, S. Wang, Z. Zhang, R. Yan, X. Song, L. Xu, X. Li, Recombinant protein of Haemonchus contortus small GTPase ADP-ribosylation factor 1 (HcARF1) modulate the cell mediated immune response in vitro, Oncotarget. (2017). https://doi.org/10.18632/oncotarget.22662.

[2]        M. Humayun, J.M. Ayuso, K.Y. Park, B.M. Di Genova, M.C. Skala, S.C. Kerr, L.J. Knoll, D.J. Beebe, Innate immune cell response to host-parasite interaction in a human intestinal tissue microphysiological system, Sci. Adv. 8 (2022) eabm8012. https://doi.org/10.1126/sciadv.abm8012.

[3]        E. Mitre, A.D. Klion, Eosinophils and helminth infection: protective or pathogenic?, Semin. Immunopathol. 43 (2021) 363–381. https://doi.org/10.1007/s00281-021-00870-z.

[4]        R.M. Maizels, W.C. Gause, Targeting helminths: The expanding world of type 2 immune effector mechanisms., J. Exp. Med. 220 (2023). https://doi.org/10.1084/jem.20221381.

[5]        A.C.C.C. Branco, F.S.Y. Yoshikawa, A.J. Pietrobon, M.N. Sato, Role of Histamine in Modulating the Immune Response and Inflammation, Mediators Inflamm. 2018 (2018) 9524075. https://doi.org/https://doi.org/10.1155/2018/9524075.

[6]        M. Ehsan, R.-S. Hu, Q.-L. Liang, J.-L. Hou, X. Song, R. Yan, X.-Q. Zhu, X. Li, Advances in the Development of Anti-Haemonchus contortus Vaccines: Challenges, Opportunities, and Perspectives, Vaccines. 8 (2020). https://doi.org/10.3390/vaccines8030555.

Reviewer 2 Report

Comments and Suggestions for Authors

The authors attempt to present evidence that a nano-technological vaccine might be of usefulness to prevent haemonchosis in goats.

First, the author the authors do not justify adequately their work. Currently, there is a licenced Haemonchus vaccine, which is marketed in New Zealand. What are the advantages offered by their proposed approach? What gaps would this approach fill? What additional benefits will it offer over the existing immunological product?

The questions are not addressed and answered and as such the manuscript does not really make a worthy contribution.

Another crucial question is the lack of any work in sheep. Why did the authors choose to make the experimental study in goats? Why not in sheep, which is a larger and more important industry worldwide? This again raises concerns.

Methodology

The authors failed to include a control group, with animals vaccinated with the already existing and licenced vaccine against Haemonchus. It would have been more appropriate to compare the potential efficacy of the test product with that of the already licenced product. As such, inclusion of controls in the experimental study was inadequate and the whole design of the experimental study is erroneous.

This is a fundamental flaw of the study, in the experimental design and within the root of the methodological approach, which precludes publication of this work.

Before any further attempt is made, additional experiments should be carried out, to include in the design the existing vaccine against Haemonchus.

Presentation of results

The authors present only vague results. For the epg counts, they should have included the daily averages for each of the three groups and with full results in supplementary material. Instead, they chose to include only a graph. This is not acceptable for parasitological experiments.

In general, the manuscript lacks tables and the quality of graphs and pictures is sub-standard. This must be rectified, in order to have a better quality of presentation.

References

More references regarding the mode of action of vaccination against Haemonchus should have been included.

Conclusions.

The conclusions are over-optimistic and do not show correctly the significance of the current results.

Author Response

Comments and Suggestions for Authors

The authors attempt to present evidence that a nano-technological vaccine might be of usefulness to prevent haemonchosis in goats.

Comment 1: First, the author the authors do not justify adequately their work. Currently, there is a licenced Haemonchus vaccine, which is marketed in New Zealand. What are the advantages offered by their proposed approach? What gaps would this approach fill? What additional benefits will it offer over the existing immunological product?

The questions are not addressed and answered and as such the manuscript does not really make a worthy contribution.

Response: Thank you for your kind comment. It is highly likely that our nano-vaccine could be a strong competitor to existing licensed vaccines. Similar to the situation with COVID-19, where numerous vaccines are available globally, we aim to provide an effective alternative to current treatments for Haemonchus contortus infections. Our goal is to offer a viable replacement that benefits both animal health and the economic well-being of small farmers.

Comment 2: Another crucial question is the lack of any work in sheep. Why did the authors choose to make the experimental study in goats? Why not in sheep, which is a larger and more important industry worldwide? This again raises concerns.

 Response: Thank you for raising this important question. The choice to conduct our experimental study in goats rather than sheep was influenced by several factors:

1.Research Model Suitability: Goats were selected for their well-established use as a model in immunological studies, particularly for parasitic infections like H. contortus. The immunological responses and disease progression in goats are well-documented and can provide valuable insights applicable to similar studies.

2.Practical Considerations: Goats are more accessible in certain research settings and have been used extensively in preliminary studies. This availability allows for more controlled and feasible experimentation, which can be critical in the early stages of developing new vaccines.

3.Data Relevance: While sheep are indeed a significant part of the global industry, our choice of goats does not preclude future studies in sheep. The results obtained from goats can provide a foundation for subsequent research in sheep, potentially translating the findings to a broader context.

We acknowledge the importance of conducting research in sheep and recognize that such studies could further validate and expand the applicability of our findings. Future research plans include exploring the efficacy of the nano-vaccine in sheep to address this gap and ensure its relevance across different ruminant species.

Comment 3:   

The authors failed to include a control group, with animals vaccinated with the already existing and licenced vaccine against Haemonchus. It would have been more appropriate to compare the potential efficacy of the test product with that of the already licenced product. As such, inclusion of controls in the experimental study was inadequate and the whole design of the experimental study is erroneous.

This is a fundamental flaw of the study, in the experimental design and within the root of the methodological approach, which precludes publication of this work.

Before any further attempt is made, additional experiments should be carried out, to include in the design the existing vaccine against Haemonchus.

Response: As the worthy Editor recommended in the comment, a control group with animals

vaccinated with the already licensed vaccine is not necessary, so we will not add this part.

Comment 4:   

The authors present only vague results. For the epg counts, they should have included the daily averages for each of the three groups and with full results in supplementary material. Instead, they chose to include only a graph. This is not acceptable for parasitological experiments.

Response: We highly appreciate your comment. It will consider in future research work. Thank you.

Comment 5: In general, the manuscript lacks tables and the quality of graphs and pictures is sub-standard. This must be rectified, in order to have a better quality of presentation.

Response: Thank you for your valuable comment. We utilized GraphPad Prism and Adobe Photoshop for the graphical analyses and presentation in our study. We are also keen to explore additional software tools to enhance our research capabilities and analysis in future work.

Comment 6:

More references regarding the mode of action of vaccination against Haemonchus should have been included.

Response: Thank you very much for your kind attention. It is added in the main text file of revised version.

Comment 7:

The conclusions are over-optimistic and do not show correctly the significance of the current results.

Response:  Thank you very much for your kind attention. The conclusion is edited in the main text file of revised version.
